# Dream Recall upon Awakening from Non-Rapid Eye Movement Sleep in Older Adults: Electrophysiological Pattern and Qualitative Features

**DOI:** 10.3390/brainsci10060343

**Published:** 2020-06-03

**Authors:** Serena Scarpelli, Aurora D’Atri, Chiara Bartolacci, Maurizio Gorgoni, Anastasia Mangiaruga, Michele Ferrara, Luigi De Gennaro

**Affiliations:** 1IRCCS Fondazione Santa Lucia, 00179 Rome, Italy; luigi.degennaro@uniroma1.it; 2Department of Psychology, Sapienza University of Rome, 00185 Rome, Italy; aurora.datri@uniroma1.it (A.D.); chiara.bartolacci@uniroma1.it (C.B.); maurizio.gorgoni@uniroma1.it (M.G.); anastasia.mangiaruga@studio.unibo.it (A.M.); 3Department of Medical and Surgical Sciences (DIMEC), University of Bologna, 40126 Bologna, Italy; 4Department of Biotechnological and Applied Clinical Sciences, University of L’Aquila, Via Vetoio (Coppito 2), 67100 Coppito (L’Aquila), Italy; michele.ferrara@univaq.it

**Keywords:** dream recall, dream report, older adults, NREM sleep, EEG, activation hypothesis, delta, beta

## Abstract

Several findings support the activation hypothesis, positing that cortical arousal promotes dream recall (DR). However, most studies have been carried out on young participants, while the electrophysiological (EEG) correlates of DR in older people are still mostly unknown. We aimed to test the activation hypothesis on 20 elders, focusing on the Non-Rapid Eye Movement (NREM) sleep stage. All the subjects underwent polysomnography, and a dream report was collected upon their awakening from NREM sleep. Nine subjects were recallers (RECs) and 11 were non-RECs (NRECs). The delta and beta EEG activity of the last 5 min and the total NREM sleep was calculated by Fast Fourier Transform. Statistical comparisons (RECs vs. NRECs) revealed no differences in the last 5 min of sleep. Significant differences were found in the total NREM sleep: the RECs showed lower delta power over the parietal areas than the NRECs. Consistently, statistical comparisons on the activation index (delta/beta power) revealed that RECs showed a higher level of arousal in the fronto-temporal and parieto-occipital regions than NRECs. Both visual vividness and dream length are positively related to the level of activation. Overall, our results are consistent with the view that dreaming and the storage of oneiric contents depend on the level of arousal during sleep, highlighting a crucial role of the temporo-parietal-occipital zone.

## 1. Introduction

Dreaming is a distinctive mental experience occurring during sleep. For years, the study of the neural bases of dreaming has been strictly linked to the Rapid Eye Movement (REM) sleep stage, while the sleep stage marked by slow waves was associated with “mental inactivity” (for a review see, [1,2,3]). When a broader definition of dreaming was introduced by Foulkes [4], it became evident that dreams can also be reported upon Non-REM (NREM) awakening, decreeing the fall of the REM dreaming equation [5]. In this vein, many investigations were conducted to discover the electrophysiological/electroencephalographic (EEG) correlates of dream experience also outside the REM sleep stage [6,7,8,9,10,11,12,13]. Although the studies reported some remarkable differences concerning the sleep EEG correlates of dream experience, it should be noted that several recent findings converge towards the so-called “activation hypothesis/model” [9,10,11,12,13], according to which a desynchronized sleep EEG would promote a dream experience [14]. In keeping with this view, results from clinical samples revealed that increased light sleep and sleep fragmentation are related to a higher dream recall frequency [15].

Among the EEG studies, except for some results that revealed alterations in alpha power at 8–10 Hz (e.g., [6,7,8]), the majority of studies point out that decreased delta power (0.5–4 Hz) predicts dream recall (DR) after awakening from NREM sleep [9,10,11,16].

It is worth noting that dream research is almost totally restricted to young participants, and we know very little about the evolution of these mechanisms during the lifespan (for a review, see [17]). In particular, the effect of aging on dreaming has been poorly investigated, despite the close relationship between aging and sleep changes [18,19]. It is well known that significant variations characterize the sleep macrostructure in older people. Specifically, total sleep time, sleep efficiency, and the amount of slow-wave sleep (SWS) are reduced [18]. Instead, intra-sleep wakefulness and day napping increase [20]. According to the relation between EEG rhythms and DR, it is possible to hypothesize that the dream experience could be affected by these age-dependent modifications of sleep.

Considering the activation hypothesis, one might expect that the sleep features in aging could promote dreaming. However, some lines of research collecting the DR frequency in older adults showed a critical age-related reduction in DR rate. On the one hand, this reduction may be associated with cognitive deterioration and difficulties in the storage and recall of dream materials [19,20,21,22]; on the other hand, it may be related to the loss of interest in one’s own mental sleep activity (i.e., dream salience) during older age [19,23,24]. However, this issue has been extensively debated since the drop in DR rate was found already during middle age [21,25], and no differences occurred between middle-aged and older adults [26].

Moreover, the qualitative and quantitative features of dream reports in older subjects are very little explored. Some studies revealed a shorter dream length in older adults than in younger [27]. Conversely, other investigations found no differences between different age ranges [28,29,30]. Concerning qualitative features, a dated study revealed a reduction in emotion in women’s reports beyond the age of 60 [26]. Moreover, to our knowledge, no studies on bizarreness or vividness have been carried out on dream reports in older people.

Beyond this, only two studies have investigated the effects of sleep EEG correlates of DR in older adults without examining dream reports. A first study was performed by multiple provoked awakenings after short sleep periods during 40 h under a constant routine [31]. The result went in the opposite direction to the activation hypothesis—DR was related to higher delta and sigma activity during NREM sleep [31]. While this investigation did not observe any differences between recallers (RECs) and non-recallers (NRECs) in REM sleep, a recent study revealed that frontal theta oscillations in older adults—as in younger [8]—predict DR upon awakening from REM sleep [32]. According to this background, firstly we aimed to investigate the EEG correlates of dream experience in older adults during NREM sleep. We hypothesized that older recallers’ EEG patterns are similar to those detected in younger recallers [9,10,11,12]. Specifically, considering the sleep features during aging, we tested the activation hypothesis, with the expectation of replicating the relationship between DR and cortical arousal. In keeping with previous studies, we should observe a higher EEG activation during pre-awakening. Second, we aimed to assess the qualitative and quantitative features of dream experience in our older sample, examining a possible relationship with sleep measures.

## 2. Materials and Methods

### 2.1. Subjects

The current study included 20 healthy older volunteers (11 F, 9 M; mean age: 68 ± 1.14 SE). Participants were recruited from local senior centers in Rome. The following inclusion criteria were considered for all subjects: the absence of psychiatric or neurological disorders and cognitive impairment; the absence of sleep disturbances and/or the condition of excessive daytime sleepiness (EDS); the absence of other relevant medical conditions and obesity; the absence of alcohol and other substance addiction; regular sleep-wake rhythms. To ascertain the eligibility for the study enrollment, a clinical and anamnestic interview was administered at the recruitment phase.

All the participants signed informed consent before the experimental session. The study protocol (#1128/2016) was approved on 1 December 2016 by the Institutional Ethics Committee of the Department of Psychology of the “Sapienza” University of Rome according to the Declaration of Helsinki.

### 2.2. Study Design

#### 2.2.1. Polysomnography

All the subjects underwent a complete polysomnographic (PSG) recording of a single night of undisturbed sleep in a sound-proof, temperature-controlled room inside the laboratory. During the week before the experimental session, they were requested to comply with a regular sleep–wake schedule.

All the PSG data were recorded by using the Micromed system plus a digital polygraph. The 19 unipolar EEG scalp derivations (Ag/AgCl electrodes) were placed following the international 10–20 system (F1, F2, F3, F4, F7, F8, Fz, C3, C4, Cz, P3, P4, Pz, O1, O2, T3, T4, T5, and T6), and the A1 and A2 from mastoids were acquired with a sampling frequency of 256 Hz and a bandpass filtered at 0.5–30 Hz. Additionally, a notch filter was applied (50 Hz). The ground electrode was placed at the frontopolar location (Fpz). In addition, electrooculogram (EOG) and submental electromyogram (EMG) signals were recorded. EOG electrodes were placed about 1 cm from the medial and lateral canthi of the dominant eye. The impedances of all electrodes were below 5 kΩ. Moreover, the variation in oxygen saturation was monitored by a pulse oximeter located on the right index finger to exclude sleep-breathing disorders.

The PSG signals of the whole night and the 5 min of stage 2 NREM sleep preceding the awakening were analog-to digital converted on-line, with a sampling rate of 256 Hz.

Bedtime was between 10:00 PM and 12:00 AM, while the morning awakening was between 5:00 and 7:00 AM depending on each participant’s usual schedule. All the subjects were awakened from stage 2 NREM sleep. The researcher checked the PSG and decided to wake the subject up when no stage shift over the last 5 min of sleep had occurred. An expert sleep scorer then confirmed the stage stability of the last segment of sleep.

#### 2.2.2. Dream Collection

Before the sleep recording, all the subjects were trained to consider all types of mental sleep activity as dream experience (both oneiric- and thought-like dreams) and how to provide their dream report upon the morning awakening. The participants were awakened in the morning by calling out their first name and were immediately invited to verbally record “everything that was going through your mind during the sleep period”, providing as many details as possible. Then, they were asked to complete a sleep and dream questionnaire [33,34,35] to collect the following information: self-reported sleep quality (sleep onset latency (SOL), total sleep time (TST), number of awakenings); the presence/absence of DR; the number of dreams during the whole night; and self-reported dream features, evaluated according to four 6-point Likert rating scales (visual vividness, emotional load, bizarreness, and length of their dream experience).

Subjects who reported at least one dream experience were defined as RECallers (RECs), while subjects who did not report any DR were defined as Non-RECs (NRECs).

### 2.3. Data Analysis

#### 2.3.1. Sleep Architecture

The PSG data was scored considering 20 sec epochs, according to the Rechtschaffen and Kales criteria [36]. SWS scoring rigidly fulfilled the >75-μV amplitude criterion. Concerning the conventional sleep parameters, the variables on sleep stages latency and duration in minutes were assessed. Additionally, the total bed time (TBT) and TST (i.e., the sum of time spent in stage 1, stage 2, SWS, and REM) were calculated. Finally, the following variables were considered indexes of sleep fragmentation: (a) wake after sleep onset (WASO) in minutes, (b) number of awakenings, (c) sleep efficiency index (sleep efficiency index (SEI) = TST/TBT × 100).

#### 2.3.2. Quantitative EEG Analysis

The EEG signals were referenced offline to the mean of the two mastoid channels (A1, A2). Concerning the 5 min of NREM stage 2 sleep, artifacts were additionally rejected off-line on an 8 s basis by visual inspection.

Both for the whole-night NREM sleep and for the last 5 min of NREM sleep, the power spectra of the 19 scalp derivations were calculated by a customized fast Fourier transform (FFT) routine in consecutive 4 sec epochs, resulting in a frequency resolution of 0.25 Hz. Values above 25 Hz were not used in the analysis. The values of EEG power from adjacent frequency bins for each scalp derivation—expressed as the percentage of the total power spectrum within the whole topography—were summed together to obtain the traditional frequency bands: delta (0.50–4.75 Hz), theta (5.00–7.75 Hz), alpha (8.00–11.75 Hz), sigma (12.00–15.75 Hz), and beta (16.00–24.75 Hz).

#### 2.3.3. Dream Reports

The tape recordings of each dream experience were transcribed verbatim. Firstly, an expert researcher, unaware of the study objectives, pruned the reports of the passages not closely related to the dream content (e.g., “I’m not sure, but in my opinion ...”) or repetitive content/sentences. Subsequently, two trained judges, both unaware of the study design, scored independently each dream report according to three 6-point Likert rating scale of visual vividness (VV), emotional load (EL), and bizarreness (B) [33,34,35]. The rating scales ranged from 1 (“a very small extent”) to 6 (“a very great extent”). As in previous studies [33,34,35], here the positive or negative emotional valence were not considered.

Concerning the B score, the judges evaluated (a) the bizarre elements (improbable or impossible characters; metamorphoses; improbable or impossible actions/inappropriate roles; improbable or impossible objects) and (b) the script bizarreness (physically improbable or impossible plot; logically improbable or impossible plot; plot discontinuity; improbable or impossible settings).

The VV of each dream report was attributed according to the following scores: (a) no image at all (only thinking of the object); (b) very vague and dim; (c) less vague, still dim; (d) moderately clear and vivid; (e) clear and reasonably vivid; (f) perfectly clear and as vivid as normal vision.

The inter-rater reliability of each rating scale was calculated (Cohen’s K coefficient >0.80), and the cases of differences between the two judges were consensually solved.

Furthermore, two quantitative variables were considered: the number of dreams and the report length expressed as the numbers of words of each dream report after pruning (Total Word Count(TWC)).

Additionally, the self-reported dream features assessed by the diaries were considered for further analysis.

To summarize, we considered the dependent variables: VV, EL, and B from external evaluation; the self-reported (sr_) VV, sr_EL, sr_B and length (sr_L) from diaries; the number of dreams (N); and the TWC.

#### 2.3.4. Statistical Analysis

Sleep architecture of the REC and NREC groups were compared by unpaired t-tests.

To test the activation hypothesis, only the delta and beta bands were considered for the principal analysis on the EEG data. The statistical comparisons between the REC and NREC groups were carried out by an unpaired t-test on the relative power of each scalp derivation, and separately for the total NREM sleep and the last 5 min of NREM sleep. The Bonferroni correction was applied to adjust the α-value for multiple comparisons [37,38]. We used the Bonferroni adjustment proposed by Sankoah et al. [38], since the EEG data are mutually correlated measures. Hence, we added the mean correlation between the EEG variables as a parameter. Considering the mean correlation between the dependent variables in stage 2 NREM sleep during the whole night (*r* = 0.12), the α level was adjusted to 0.002. Considering the mean correlation between the dependent variables in the last 5 min of stage 2 NREM sleep (*r* = 0.07), the α level was adjusted to 0.0017.

Bearing in mind the activation hypothesis, we also calculated the delta/beta power ratio as an integrated EEG index of activation. Previously, this index was proved to be reliable predictor of DR [13]. The comparisons of the activation index between the REC and NREC groups were carried out by unpaired t-tests for each scalp derivations and separately for the pre-awakenings and the total NREM sleep. To adjust the α level, the Bonferroni correction was applied [37,38]. Considering the mean correlation between the dependent variables (*r* = 0.92), the α level was adjusted to 0.038 for the pre-awakening comparisons. Considering the mean correlation between the dependent variables (*r* = 0.81), the α level was adjusted to 0.028 for the NREM total sleep comparisons.

Correlational analyses (Pearson’s coefficient, *r*) between the power values of the derivations showing significant REC vs. NREC differences and the dream features (from self-report and external judges) were carried out. Additionally, the correlations between the activation index and dream experience were tested.

Finally, to test if the dream reports were related to sleep fragmentation, correlations were carried out between the dream features and the WASO, number of awakenings, and SEI. The significance for the correlational analyses was set at *p* < 0.05 (two-tailed).

## 3. Results

Nine out of 20 older subjects recalled at least one dream after awakening (45% of the sample; mean number of recalled dreams = 1.67; SE = 0.24), and the other 11 subjects were NREC.

### 3.1. Sleep Architecture

Table 1 shows the results of the unpaired t-tests comparing the sleep measures in the REC and NREC groups. No differences were found in the sleep architecture between the two conditions.

### 3.2. EEG Pattern of Dream Recall: Last 5 min of NREM Sleep

Figure 1 shows the topographic distribution of the mean relative EEG power in the delta and beta frequency bands for the last 5 min of NREM sleep in the REC and NREC groups and the statistical maps of the comparisons between groups (assessed by an unpaired *t*-test). It should be noted that the EEG regional patterns are substantially preserved in both the REC and NREC groups. Statistical comparisons revealed no significant differences in delta and beta power between the RECs and NRECs during the last 5 min of NREM sleep. To ascertain that no difference was present in other bands (theta, alpha, and sigma), we carried out supplementary analyses (please see Appendix A).

### 3.3. EEG Pattern of Dream Recall: Total NREM Sleep

Figure 2 shows the topographic distribution of the mean relative EEG power in the delta and beta frequency bands for the total NREM sleep of the REC and NREC groups and the statistical maps of the comparisons between groups (assessed by an unpaired t-test). As well as for the last 5 min, the distributions show that the EEG regional patterns are substantially preserved in both the REC and NREC groups. Statistical comparisons revealed significant differences in the correspondence of the temporo-parietal area: RECs show lower delta power in correspondence to Pz (t = −355; *p* = 0.002) and T6 (t = 4.07; *p* = 0.0007), as compared to the NRECs. Moreover, the REC group shows a higher beta power over the frontal-temporal areas than the NREC group; nevertheless, these comparisons are not statistically significant after the Bonferroni correction.

To ascertain that no difference was present in other bands (theta, alpha, and sigma), we carried out supplementary analyses (please see Appendix A).

### 3.4. Activation Index as Predictor of Dream Recall

Figure 3 shows the comparisons (assessed by unpaired *t*-tests) of the activation index in the REC and NREC groups. During the pre-awakening, no comparison was significant. Differently, the results showed that DR is related with a higher level of cortical activation—namely, lower values of the activation index—during the whole night compared with the absence of dream report. Specifically, differences were significant in 9 of 19 derivations: F7 (*t* = −2.42; *p* = 0.026), F8 (*t* = −2.53; *p* = 0.021), O1 (*t* = −3.08; *p* = 0.006), O2 (*t* = −2.61; *p* = 0.018), P4 (*t* = −2.75; *p* = 0.013), Pz (*t* = −2.65; *p* = 0.016), T4 (*t* = −2.88; *p* = 0.01), T5 (*t* = −2.50; *p* = 0.022), T6 (*t* = −3.26; *p* = 0.004).

### 3.5. Sleep Pattern and Dream Features

Table 2 shows the mean and standard errors of each dream feature examined.

Correlations were carried out between the derivations in which differences between the RECs and NRECs are significant (Pz and T6) for the delta power and all the dream features considered. Results reveal that the both VV and sr_VV negatively correlate with delta power on Pz, respectively, with *r* = −0.72 (*p* = 0.028) and *r* = −0.69 (*p* = 0.039). For illustrative purposes, Appendix A shows the topographic distribution of the Pearson’s *r* coefficients of the correlations between the delta activity and the VV.

Moreover, the correlations were performed between the derivations in which we detected significant differences concerning the activation index. The results revealed that the level of activation in the parieto-occipital area is associated with the self-reported length (sr_L) of DR: O1 (*r* = −0.68; *p* = 0.043), P4 (*r* = −0.84; *p* = 0.004), Pz (*r* = −0.86; *p* = 0.003). Specifically, higher values of the activation index (i.e., lower arousal) are associated with a lower sr_L. For illustrative purposes, Appendix A shows the topographic distribution of the Pearson’s *r* coefficients of the correlations between the activation index values and the sr_L of dream reports.

Finally, the correlations between the macrostructural indexes of sleep fragmentation and dream features revealed that: a) WASO positively correlates with the VV (*r* = 0.70; *p* = 0.036); b) SEI negatively correlates with sr_L (*r* = −0.71; *p* = 0.034).

## 4. Discussion

Here, we tested the activation hypothesis on older individuals upon awakening from NREM sleep. The lack of intraindividual measures and the relatively small sample size make the present results preliminary, and future studies performed in larger samples will be required to ultimately confirm their validity.

Consistently with previous studies on young adults [9,10,11,12,13], we reported that elders recall their mental sleep activity after awakening when the electrophysiological milieu during their entire nocturnal NREM sleep is characterized by a lower delta power in the temporo-parietal area. Contrary to our expectation, we did not find any differences between RECs and NRECs in the last segment of sleep (i.e., pre-awakening). Nevertheless, we observed a similar topographic pattern in the delta band (i.e., lower delta over the parietal area). The delta/beta activation index highlighted the same effect for the entire sleep: the RECs during NREM sleep of the whole night reported higher arousal levels than the NRECs. The activation is spatially diffuse, with significant differences over the fronto-temporal and parieto-occipital areas.

Our findings are consistent with the topographic EEG correlates of DR upon awakening from NREM sleep reported by previous studies. Indeed, the decrease in the power of the delta band was found to correspond with the so-called “hot zone” [10,39]. Specifically, the localization in the posterior areas of fast frequency (>20 Hz) and low frequency EEG activity related to dreaming during NREM sleep has been observed in healthy subjects [10]. Similarly, narcoleptics who recalled their dream experience showed a higher beta power and reduced delta power over centro-parietal areas during naps [13]. It is well known that parietal areas are essential during wakefulness for visuospatial skills [40]. Additionally, the temporo-parietal junction is involved in the cortical network responsible for mental imagery and visual memory [41]. Interestingly, some studies on DR in children revealed a relationship between dreaming and visuospatial /visual memory skills, which depend on the brain maturation of posterior areas (for a review, see [17,19]). Additionally, studies on the complete cessation of dreaming showed the pivotal role of parietal and occipital areas in generating and recalling the processes of dream experience [42,43,44,45].

However, we have to underline that we found significant differences in older adults only during the whole night. It is worth noting that the negative finding on the last 5 min of sleep could be ascribed to our choice not to ask about the mental sleep activity “just before the awakenings” but referring to the entire sleep period. It could be hypothesized that the phrasing of that question could explain why we failed to find any differences in the pre-awakening period.

The larger effect on total NREM sleep than the last 5 min of sleep suggests that for elders, lighter sleep and higher desynchronization during the night is necessary to consolidate the memory trace of their dream experience. This appears particularly consistent with the arousal-retrieval model [46,47]; a certain arousal level and intrasleep waking periods are a prerequisite to encode the dream experience and ensure successful DR in the morning.

Moreover, the fact that DR may be related to the course of the whole night is not new. For instance, after a recovery night (i.e., a night dominated by a higher amount of SWS), sleep-deprived subjects have a significant drop in their DR rate [48]. Similarly, individuals suffering from sleep apnea treated with Continuous Positive Airway Pressure reported a low DR rate along with deeper sleep [49]. Additionally, a study on patients with central hypersomnia reporting an absence of experiences or recall of them from sleep onset to offset (i.e., night blackout) revealed consistent findings: subjects who experienced a night blackout had a higher SWS, and a greater percentage of these reported lower DR frequency [50]. In line with this evidence, the insomnia condition—characterized by higher activation [51]—appears to be related to a high DR rate [15].

For the first time, the current study investigated the relationship between EEG patterns and dream features in older people. Although we do not have a direct comparison with a younger sample, we estimated that 45% of our older adults remembered at least one dream after NREM sleep awakening. We might have expected that sleep inertia—which is greater after NREM than REM sleep [52,53,54]—can affect the possibility of recalling the dream experience in older adults. However, it should be noted that the DR percentage after NREM sleep is only slightly lower compared to a previous study in which older adults were awoken from REM sleep (57% were RECs) [32].

We also observed a relationship between the visual vividness (both VV and sr_VV scores) of dream reports and lower delta activity corresponding to centro-parietal areas. Once again, this finding suggests that slow oscillations/delta waves impact on the dream experience and, specifically, on its visual component. Besides this, the correlations between dream features and the activation index showed that a higher level of arousal with fronto-temporal and posterior locations is associated with the feeling of having longer dreams. Moreover, VV is also related to greater intrasleep wakefulness, and poor sleep quality—expressed in terms of SEI—is associated with the feeling of having longer dreams. Overall, these findings are consistent with the activation hypothesis claiming that dream features depend on the periodic and distributed level of arousal during sleep [14], in particular over the temporo-parietal-occipital zone, supporting perceptual and visual processes during wakefulness.

## 5. Limitations

This study has some limitations that should be highlighted. Several of these are intrinsic to the specific sample examined, since the elders showed a lack of willingness to change their routines and spend the night in a laboratory:(a)the absence of the adaptation night that could have affected the continuity of sleep during the night and does not allow us to rule out the so-called first night effect;(b)the lack of intraindividual repeated measures and a relatively small sample size, not allowing us to disentangle the state-/trait-like issue [2].

Besides this, a comparison with a younger sample is missing. We suggest that future studies should provide comparisons between older and young adults, especially concerning the features of dream experiences.

Finally, we have to mention a limitation due to the data recording: the EEG filtering set at 30 Hz did not let us acquire and analyze the gamma band. Interestingly, some studies demonstrated that a higher gamma power (a) predicts DR both during REM and NREM sleep [10]; (b) is related with lucid dreams [55,56]; and (c) is an EEG marker of the frequent production of nightmares [57].

## 6. Conclusions

On the one hand, our results suggest that a high delta power during sleep excludes the possibility of having a conscious experience. On the other hand, leaving aside the issue of the “production” of dreaming and considering that DR is an episodic memory trace, we could posit that light sleep and EEG desynchronization in elders promote the storage of mental sleep activity and its recall after awakening. Interestingly, we highlighted that a more activated EEG milieu and a greater sleep fragmentation also significantly impact on dream features.

To sum up, our findings are consistent with the activation hypothesis, and the EEG correlates of dream experience in older adults seem to be partially comparable to those observed in younger samples.

## Figures and Tables

**Figure 1 brainsci-10-00343-f001:**
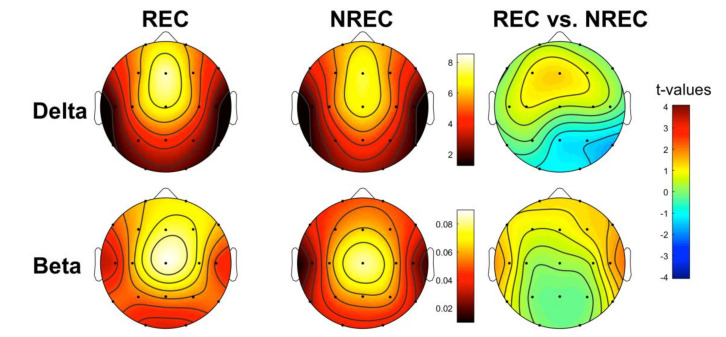
Topographic distribution of the last 5 min of NREM sleep electrophysiological (EEG) power in the non-recall (NREC) and recall (REC) groups and statistical maps. The EEG activity for the delta (0.5–4.75 Hz) and beta (16–24.75 Hz) frequency bands is reported and expressed as the percentage of the total EEG power within the whole topography. The maps are scaled between the minimal and maximal power values for each frequency band, considering both the NREC (1st column) and REC (2nd column) groups. Statistical maps of the comparisons (unpaired *t*-test) between the NREC and REC groups are also plotted (3rd column) both for the delta and beta band. The statistical maps are scaled symmetrically according to the absolute maximal *t*-value across the statistical comparisons. No significant differences were found (*p* ≤ 0.0017).

**Figure 2 brainsci-10-00343-f002:**
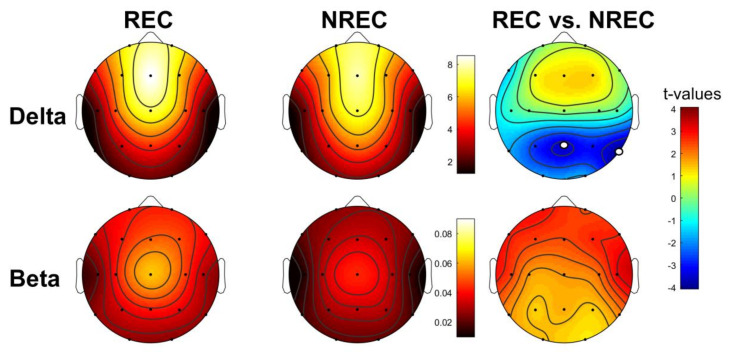
Topographic distribution of the total NREM sleep EEG power in the non-recall (NREC) and recall (REC) groups and statistical maps. The EEG activity for the delta (0.5–4.75 Hz) and beta (16–24.75 Hz) frequency bands is reported and expressed as the percentage of the total EEG power within the whole topography. The maps are scaled between the minimal and maximal power values for each frequency band, considering both the NREC (1st column) and REC (2nd column) group. Statistical maps of the comparisons (unpaired *t*-test) between the NREC and REC groups are also plotted (3rd column) both for the delta and beta bands. The statistical maps are scaled symmetrically according to the absolute maximal *t*-value across the statistical comparisons. Significant differences after the Bonferroni correction for multiple comparisons are indicated by white dots (*p* ≤ 0.002).

**Figure 3 brainsci-10-00343-f003:**
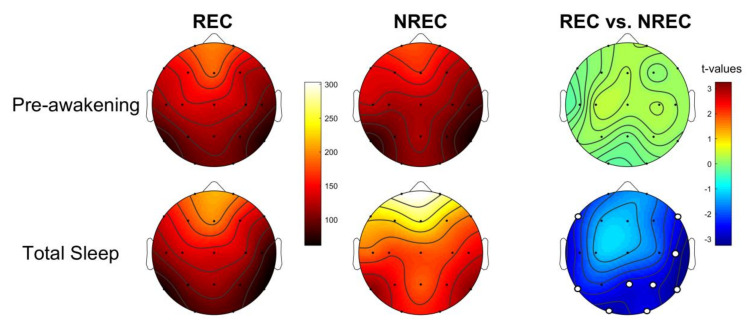
**Activation index.** Topographic distribution of the activation index, defined as the delta/beta ratio, for the last 5 min of NREM sleep (1st row) and the total NREM sleep (2nd row) in the non-recall (NREC, 1st column) and the recall (REC, 2nd column) groups. The maps are scaled between the minimal and maximal, considering both groups. Statistical maps of the comparisons (unpaired *t*-test) between the REC and NREC groups are also plotted (3rd column) both for the pre-awakening and the total NREM sleep. The statistical maps are scaled symmetrically according to the absolute maximal *t*-value across the statistical comparisons. Significant differences after the Bonferroni correction for multiple comparisons are found only for the total NREM sleep and are indicated by white dots (*p* ≤ 0.028).

**Table 1 brainsci-10-00343-t001:** Means and standard errors (SEs) of the polysomnographic (PSG) variables of the recallers (REC) and non-recallers (NREC) groups. The results of the unpaired *t*-test are also reported. REM, rapid eye movement; WASO, wake after sleep onset; TST, total sleep time; TBT, total bed time; SEI, sleep efficiency index; #, number.

	REC		NREC		*t-Values*	*p-Values*
	Mean	SE	Mean	SE		
**Stage 1 latency (min)**	47.41	15.21	28.21	9.87	1.10	0.29
**Stage 2 latency (min)**	20.25	8.29	17.12	4.30	0.35	0.73
**Stage 3 latency (min)**	58.06	15.35	57.48	10.15	0.03	0.98
**REM sleep latency (min)**	132	26.14	88.72	13.18	1.56	0.14
**Stage 1 (min)**	14.45	3.29	14.36	2.86	0.02	0.99
**Stage 2 (min)**	235.11	11.38	244.21	11.14	−0.57	0.58
**SWS (min)**	1.33	0.66	3.73	1.80	−1.15	0.27
**REM (min)**	45.33	7.04	51.73	4.72	−0.78	0.26
**WASO (min)**	72.25	16.58	53.75	9.54	1.01	0.33
**Awakenings (#)**	19.44	2.82	14.18	2.32	1.45	0.16
**TST (min)**	282.40	24.03	305.62	9.28	−0.97	0.34
**TBT (min)**	388.70	13.94	374.18	10.87	0.83	0.42
**SEI% (TST/TBT)**	76.5	0.04	81.4	0.03	−1.01	0.33

**Table 2 brainsci-10-00343-t002:** Means and standard errors (SEs) of the dream features. # DR, number of dreams; Sr_EL, self-reported emotional load; Sr_VV, self-reported visual vividness; Sr_B, self-reported bizarreness; Sr_L, self-reported length; EL, emotional load; VV, visual vividness; B, bizarreness; TWC, total word count, #, number.

**Dream features**		**Mean**	**SE**
	**# DR**	1.67	0.24
	**Sr_EL**	3	0.71
	**Sr_VV**	3.33	0.76
	**Sr_B**	2.56	0.69
	**Sr_L**	2.11	0.59
	**EL**	1.89	0.39
	**VV**	4.22	0.32
	**B**	2.67	0.60
	**TWC**	83.78	41.48

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
