# Peer review of "Dream Recall upon Awakening from Non-Rapid Eye Movement Sleep in Older Adults: Electrophysiological Pattern and Qualitative Features"

_brainsci, 2020, doi:10.3390/brainsci10060343_

Round 1

Reviewer 1 Report

The present cross-sectional study investigates dream recall frequency and qualitative characteristics of the dreams upon awakening from stage 2 NREM sleep, with a major focus on the associated NREM sleep EEG topography of brain activation patterns and standard overnight sleep PSG variables, in a sample of twenty 68-yrs-old healthy volunteers. Based on careful psychological evaluation and stringent criteria, 9 participants were classified as dream recallers (RECs) and 11 subjects as non-dream recallers (NRECs), respectively. Next, using FFT quantitative EEG analyses, the topographic brain activation patterns derived from the total amount of overnight NREM sleep and from the last 5-min NREM sleep portion before awakening were compared between the RECs and NRECs groups. The main research question addressed in this study was to test the brain activation hypothesis of dreaming production and dream recall during aging. In addition, the study aimed to further explore qualitative features of dream experience with aging in relation to specific sleep PSG/EEG measures in older subjects. The literature review is sufficient and appropriate to argue the study goals and formulated hypotheses logically. The methodologies applied are ample and are given in sufficient detail, thus guaranteeing obtaining of reliable results. The results are presented clearly within text and in the items and supplementary figures. They are discussed in sufficient depth and support the conclusions. Several study limitations are acknowledged correctly in the discussion section. The manuscript is presented in intelligible and scholar fashion.

I enjoyed very much reading this article. The overall pattern of obtained results is interesting and novel. It corresponds well to the study purposes and presents new insights the sleeping brain activation patterns during aging that may underlie dream recall and oneiric features in old subjects. More specifically, the findings that show topographic distribution of delta and beta EEG power and the activation index (delta/beta ratio) in association with dream recall and oneiric plot are of particular interest. Another study advantage is that some of the obtained here results also appear consistent with the arousal-retrieval model of dream recall. To the best of my knowledge, the current study explores the relationship between sleep EEG patterns and dream features in old subjects for the first time. I think therefore that the manuscript presents one timely and advantageous study, which also has the potential to better elucidate some sleep-dependent mechanisms proposed to produce a range of cognitive and psychological problems frequently observable during aging.

Author Response

We thank the reviewer for appreciating our work and supporting it for publication.

Reviewer 2 Report

In this manuscript the Authors extend, for the extent that is possible, the scenario supporting the activation-hypotesis of cortical arousal -largely documented in young patients- also to older people (they consider a random sample -of populosity twenty- that is a collection of people around 70 years old).

The research is properly documented, all the stages of the experiments have been carefully performed and properly summarized in the paper whose readability is high.

Statistics has been taken with proper care (e.g. the correct via Bonferroni adjustment their dealing with correlates) and results are reasonable.

Further, I appreciated that the Authors clearly write positive conclusions on the carried study -fairly confirming (for the significancy that such a small number of involved persons has) the "young scenario" also in elderly- as well as limitations of their approach (ultimately related to the small -and delicate- cluster of target elements to analyze).

The lenght of the paper is sound, topographic images well defined, inferred values as reported in the tables sound and the resarch seems overall well calibrated: I enjoyed reading this paper.

I think that after a careful revision to check any possible typo and, eventually, taking care of the supplementary material** the paper can be published.

**I did not undertand why your supplementary material is just those three images (can't them be simply merged within the main text)? Or maybe you may want to add a few words if you keep them as a supplementary material (I mean, at present -if I'm not wrong- these are "just 3 bare -and very nice and informative- images...).

Author Response

We thank the reviewer for his/her comments and appreciation. We have checked for typos and grammar errors as suggested.

Also, according to the suggestions, we merged the three images into a single supplementary file, including short comments for each figure.

Reviewer 3 Report

In their study, Scarpelli and colleagues investigated the neural correlates of NREM-sleep conscious experiences in a group of older adults. They found that dream recall, relative to no dream recall, was associated with lower delta/beta power ratio ("activation index") as measured during NREM-sleep of the whole night. In addition, the authors found significant correlations between the activation index and dream properties including visual vivdness and dream length. Similar analyses in the 5 min preceding the awakening yeilded no significant results.

The study is timely and interesting. The experimental approach and the data-analysis methods are sound. The results of this work extend and complement previous findings obtained in younger adult subjects.

The main limitations of this work are the lack of intraindividual repeated measures and the relatively small sample size. Both these aspects are adequately discussed by the authors. In addition to highlight these limitations, however, the authors should explicitly state in the Discussion section that, due to these issues, the present results should be considered as preliminary and that future studies performed in larger samples will be required to ultimately confirm their validity.

Below are my additional comments and suggestions for the authors:

Line 129: "Participants were awakened in the morning by calling out their first name and were immediately invited to verbally record “everything that was going through your mind during the sleep period".” I understand the choice of the authors to use this phrasing instead of the more common "just before the alarm sound" (or "just before you woke-up") given that only one dream report was collected in the morning. However, in response to this question, participants may have described experiences that they had during the night and not just right before they woke-up. This may explain why the authors failed to find any differences in the pre-awakening period. The authors should discuss this issue as a possibile limitation for the pre-awakening analysis.

Line 134: "Self-reported dream features evaluated according to three 6-point Likert rating scales [visual vividness, emotional load, bizarreness, and length of their dream experience]." These look like four scales.

Line 131: "Then, they were asked to complete a sleep and dream diary [33-35] to collect the following information [...]". If I understood correctly, here the authors refer to something that the participants completed only once, upone awakening in the morning. If correct, the use of the term "diary" may generate some confusion since readers may associate this term to something that is done to keep track of events across multiple days. I would replace the term "diary" with "questionnaire".

Line 200: "To adjust the α level Bonferroni correction was apply [37-38]." The word "apply" should be replaced with "applied".

Line 218: "we observed a trend concerning the WASO 218 (t=1.45; p=0.16), showing that RECs tend to have a greater wake during the night, compared to NRECs." This statistics is not consistent with the one reported in Table 1, for which a p=0.16 is reported for the number of awakenings, not WASO. More importantly, however, a p-value of 0.16 should not be defined as a trend, as this term commonly indicates a p-value between 0.05 and 0.1. In fact, a p-value of 0.16 is quite far from significance and the authors should thus consider rephrasing the relative section of the results or entirely eliminating the comments regarding this "non-result".

Line 224: "maps of the between groups comparisons (assessed by unpaired t-test) between". And also line 232: "the between groups comparisons (assessed by unpaired t-test) between". The additional "between" ad the end of the sentences should be removed.

Line 253: "reveal that the both VV and sf_VV negatively correlate with delta power on Pz, respectively with r= -0.72 (p=0.028) and r= -0.69 (p=0.0039)". Something seems wrong with these p-values. An r of 0.69 should be associated with a higher p-value with respect to an r of 0.72.

Line 266: "related with higher TWC, although the correlation does not reach statistical significance (r=0.56; p=0.11)." Similar to my previous comment, this p-value does not even indicate a trend. In my opinion this correlation should not be indicated as a tendency, althouth, of note, it would be absolutely correct to report it as a null result.

Author Response

The main limitations of this work are the lack of intraindividual repeated measures and the relatively small sample size. Both these aspects are adequately discussed by the authors. In addition to highlight these limitations, however, the authors should explicitly state in the Discussion section that, due to these issues, the present results should be considered as preliminary and that future studies performed in larger samples will be required to ultimately confirm their validity.

We thank the reviewer for his/her comment. The reviewer is right and now we mention this issue in the discussion section.

Below are my additional comments and suggestions for the authors:

Line 129: "Participants were awakened in the morning by calling out their first name and were immediately invited to verbally record “everything that was going through your mind during the sleep period".” I understand the choice of the authors to use this phrasing instead of the more common "just before the alarm sound" (or "just before you woke-up") given that only one dream report was collected in the morning. However, in response to this question, participants may have described experiences that they had during the night and not just right before they woke-up. This may explain why the authors failed to find any differences in the pre-awakening period. The authors should discuss this issue as a possibile limitation for the pre-awakening analysis.

We thank the reviewer for his/her comment. We have now mentioned this crucial issue in the discussion section.

Line 134: "Self-reported dream features evaluated according to three 6-point Likert rating scales [visual vividness, emotional load, bizarreness, and length of their dream experience]." These look like four scales.

The typo has been corrected.

Line 131: "Then, they were asked to complete a sleep and dream diary [33-35] to collect the following information [...]". If I understood correctly, here the authors refer to something that the participants completed only once, upone awakening in the morning. If correct, the use of the term "diary" may generate some confusion since readers may associate this term to something that is done to keep track of events across multiple days. I would replace the term "diary" with "questionnaire".

We thank the reviewer for his/her suggestion. We have changed the word in the manuscript.

Line 200: "To adjust the α level Bonferroni correction was apply [37-38]." The word "apply" should be replaced with "applied".

The reviewer is right. The sentence has been corrected.

Line 218: "we observed a trend concerning the WASO 218 (t=1.45; p=0.16), showing that RECs tend to have a greater wake during the night, compared to NRECs." This statistics is not consistent with the one reported in Table 1, for which a p=0.16 is reported for the number of awakenings, not WASO. More importantly, however, a p-value of 0.16 should not be defined as a trend, as this term commonly indicates a p-value between 0.05 and 0.1. In fact, a p-value of 0.16 is quite far from significance and the authors should thus consider rephrasing the relative section of the results or entirely eliminating the comments regarding this "non-result".

Many thanks for the advice. The relative section of the results (and discussion) has been removed.

Line 224: "maps of the between groups comparisons (assessed by unpaired t-test) between". And also line 232: "the between groups comparisons (assessed by unpaired t-test) between". The additional "between" ad the end of the sentences should be removed.

Many thanks for the advice. The sentences have been corrected.

Line 253: "reveal that the both VV and sf_VV negatively correlate with delta power on Pz, respectively with r= -0.72 (p=0.028) and r= -0.69 (p=0.0039)". Something seems wrong with these p-values. An r of 0.69 should be associated with a higher p-value with respect to an r of 0.72.

Thanks for the advice. The typo has been corrected (p =0.039).

Line 266: "related with higher TWC, although the correlation does not reach statistical significance (r=0.56; p=0.11)." Similar to my previous comment, this p-value does not even indicate a trend. In my opinion this correlation should not be indicated as a tendency, althouth, of note, it would be absolutely correct to report it as a null result.

Again, the reviewer is right. The relative section of the results has been removed.